# High Consumption of Discretionary Beverages in Young Australian Adults Aged 18–30 Years: A Cross-Sectional Study

Alana Duncan [1,2,*], Anna Rangan [1,2], Pui Ying Ho [1,2], Virginia Chan [1,2], Alyse J. Davies [1,2], Lyndal Wellard-Cole [3] and Margaret Allman-Farinelli [1,2]

1   Discipline of Nutrition and Dietetics, Susan Wakil School of Nursing and Midwifery, Faculty of Medicine and Health, The University of Sydney, Sydney, NSW 2006, Australia
2   Charles Perkins Centre, The University of Sydney, Sydney, NSW 2006, Australia
3   Cancer Prevention and Advocacy Division, Cancer Council NSW, Woolloomooloo, NSW 2011, Australia
*   Correspondence: adun2916@alumni.sydney.edu.au

**Abstract:** Despite health advice and campaigns, discretionary beverages remain a source of added sweeteners (sugar and intense sweeteners) and fat in the dietary intakes of many young adults. This study aimed to determine discretionary beverage consumption amongst 18 to 30-year-olds residing in New South Wales, Australia. Data were collected in 2017/2018 during the MYMeals study in which 1044 participants recorded their food and beverage consumption over a three-day period, using the purpose-designed Eat and Track (EaT) app. Discretionary beverages included all water-based and milk-based drinks with added sugar, intense sweeteners or fats and excluded alcoholic beverages. Descriptive statistics were used to analyse the proportion of consumers for different types of beverages, and contribution to overall energy and nutrient intakes. ANCOVA analyses compared the energy and nutrient intakes of consumers and non-consumers, adjusted for gender and age group. Sixty-two percent of participants with complete data ($n = 1001$) were classified as consumers of discretionary beverages. The most consumed beverages were soft drinks (39.0%) and flavoured tea/coffee (23.1%). The greatest proportion of nutrients contributed by discretionary beverages was total sugars (27.2% of total per consumers). In comparison to non-consumers, consumers of discretionary beverages had higher mean daily intakes of energy (kJ) (8736 versus 7294), and higher percentage energy (%E) from total sugars (16.5 versus 13.3) ($p < 0.001$) and saturated fat (12.5 versus 12.0) ($p < 0.05$) but lower protein (18.5 versus 20.5) ($p < 0.001$). The consumption of non-alcoholic discretionary beverages continues to be a source of significant energy and total sugars among young adults.

**Keywords:** discretionary beverages; non-alcoholic discretionary beverages; sugar-sweetened beverages; soft drink; nutrition; young adults; diet record application



## 1. Introduction

It is evident that overweight and obesity is an ongoing global epidemic [1–4]. In Australia, more than two-thirds of the adult population is overweight or obese [4–6]. The rapid weight gain in young adults is of concern [3,4,7]. According to national statistics, 39% of 18–24-year-olds were overweight or obese in 2014–2015, whereas, in 2017–2018, the prevalence had risen to 46% [5]. Research has suggested that consumption of beverages with added sugar contributes to overweight and obesity [1,2,4,6,8–15]. High added sugar content in beverages has a low satiety effect, thus consumption may result in higher total energy intake because there is little compensation for energy from liquids compared with solids [1,2,11,13].

Studies have shown that excessive consumption of discretionary beverages, in particular sugar-sweetened beverages (SSB), may also lead to several other health burdens [1,2,4,6,9–11,13–16]. As glucose rapidly increases plasma insulin, excessive

discretionary beverage intake may be associated with the risk of Type 2 Diabetes Mellitus [1]. Excessive consumption of discretionary beverages may also result in dental caries [6,10,11,13–15], and the metabolism of sugar by plaque bacteria may cause demineralisation and erosion of teeth [10]. The presence of caffeine, particularly in energy drinks, may be responsible for rising systolic and diastolic blood pressure, increasing platelet aggregation and impairing endothelial function [16]. Therefore, consumption of discretionary beverages may also heighten the risk of cardiovascular disease [1,2,6,9,14–16]. More recently, beverages with intense sweeteners (very low or no energy containing sweeteners used to replace sugar) have been linked with adverse health effects, in particular changes in mouse microbiome that may impact cardiometabolic health [17]. However, overall findings for discretionary beverages are inconclusive and more research is needed [1,17].

Previous research has indicated that discretionary beverage intake amongst young Australian adults is a significant concern [8,12,18,19]. Discretionary beverage intake is defined as both sugar-sweetened beverages and beverages sweetened with intense sweeteners and is inclusive of carbonated beverages, cordials and flavoured mineral waters. Fruit and vegetable drinks that contain some juice but mostly comprise added water, sweetener and preservatives are included. According to the 2017–2018 National Health Survey (NHS), 56% of 18–34-year-olds consumed discretionary beverages at least once weekly and 12% reported a daily intake [18]. The 2011–2012 National Nutrition and Physical Activity Survey (NNPAS), the most recent nationally representative dietary survey, suggested that 54% of 19–30-year-olds consumed discretionary beverages on the day of the survey [19]. The most consumed discretionary beverages amongst this age group included soft drinks and flavoured mineral water (40%), followed by fruit and vegetable drinks (14%) [19].

A comprehensive study on discretionary beverage intake amongst young adults in Australia has not been reported since the publication of the 2011–2012 NNPAS. The aim of this research was to provide updated evidence for 18–30-year-olds regarding non-alcoholic discretionary beverage consumption inclusive of sweetened water-based beverages and milk-based-beverages that are sweetened and may have added syrups and fats such as cream. The age groups 18–24 and 25–30 have been distinguished, as it was hypothesised that the older group may have lower consumption of discretionary beverages. Furthermore, a comparison of intakes by gender will be investigated as sources of discretionary beverages may be gender specific [19]. Additionally, this research determines the contribution of discretionary beverages to overall energy and nutrient intake, and examines differences in energy and nutrient intakes between consumers and non-consumers of discretionary beverages.

## 2. Materials and Methods

### 2.1. Study Population

This project utilised the data from a cross-sectional study known as the MYMeals study, the protocol for which has previously been published [3,7,20]. For participants to be eligible, participants needed to meet the following inclusion criteria: (1) aged 18–30 years; (2) English-speaking; (3) an owner of a smart phone and (4) consume at least one beverage, meal or snack outside of home per week [7,20]. Participants were excluded if they: (1) failed to meet the inclusion criteria; (2) were not able to record dietary intake over a three-day period; (3) had ever been diagnosed with an eating disorder and/or (4) were pregnant or breastfeeding [7,20]. See Figure 1 for a flow chart of the study design. The study was approved by the University of Sydney Human Research Ethics Committee (project 2016/546).

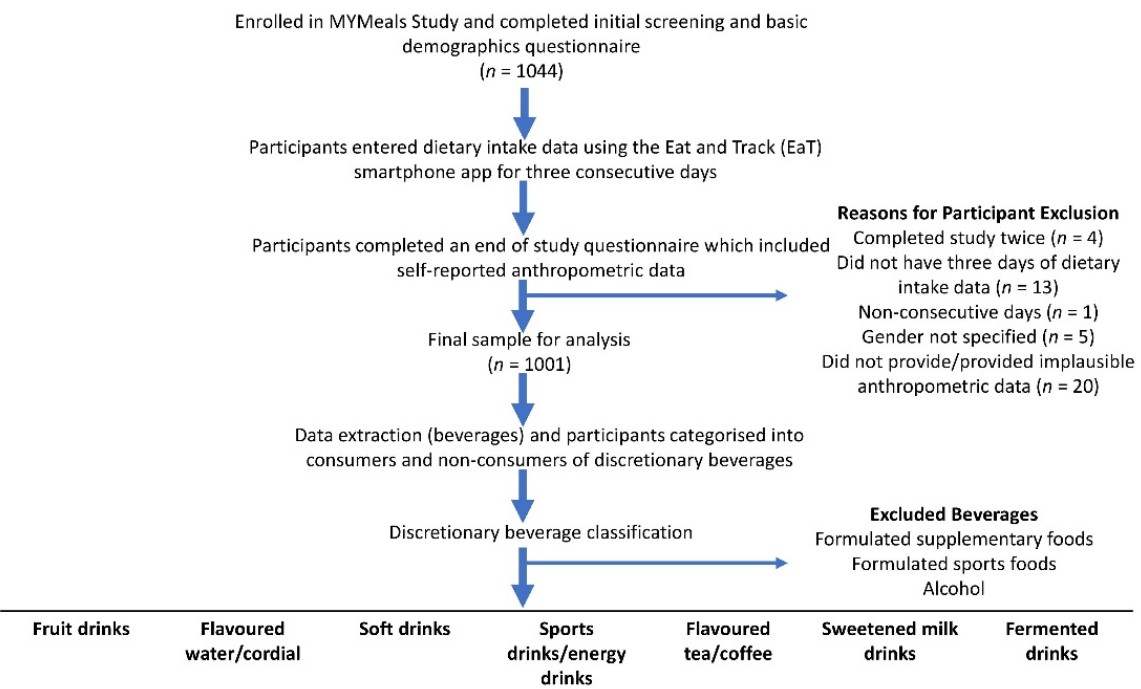

**Figure 1.** Flow diagram of the study design and discretionary beverage categorisation.

## 2.2. Data Collection

To collect dietary intake data, all participants used the Eat and Track (EaT) smartphone app [7,20]. Data were collected from April 2017 to December 2018. This involved the recording of all food and beverages consumed by participants over three consecutive days [7,20]. The nutrition database comprised of 4046 foods and beverages from the AUSNUT 2011–2013 dataset and 2229 food items from the largest food chain outlets in Australia [3,7,20]. To determine the amount of food and beverage ingested, the EaT app provided household measurements (e.g., cups, teaspoons and tablespoons) and metric measurements (e.g., grams and millilitres) [7,20]. If the food or beverage was unavailable in the app database selection, participants were able to enter the data (ingredients and amounts consumed) manually [3,7,20]. The EaT app also included questions regarding the eating occasion (i.e., breakfast, lunch, dinner, snack and beverage) and the location of food preparation (i.e., home) [7,20]. Two Accredited Practising Dietitians checked all app entries separately for any discrepancies [7]. Errors with data entries (e.g., skipped meals and incorrect unit sizes) were resolved by contacting the participants [7,20].

Participants self-reported age group (18 to 24 years or 25 to 30 years), gender (male, female, prefer not to say), weight (kg) and height (cm), highest qualification (high school, trade or diploma, university degree) and their residential postcode to determine relative socioeconomic advantage and disadvantage ranking within Australia (high; top 5 deciles or low; bottom 5 deciles) [21,22]. Body Mass Index (BMI) was calculated as weight (kg)/height (m$^2$).

## 2.3. Data Analysis

The categorisation of non-alcoholic discretionary beverages was as follows: (1) fruit drinks (i.e., <100% fruit juice with added sugar); (2) flavoured water/cordial; (3) soft drinks i.e., carbonated drinks, consumed on their own or as part of an alcoholic mixed drink; (4) sports drinks/energy drinks; (5) flavoured tea/coffee with added sugar and fats (i.e., syrups, cream); (6) sweetened milk drinks with added fat (e.g., milkshakes, thickshakes) and (7) fermented drinks (e.g., kombucha). To be classified as sweetened, the beverage needed to contain either added sugar (e.g., sugar, honey, syrups), an intense sweetener, or added saturated fat (e.g., cream, ice-cream). However, there were some exceptions to this inclusion criterion as our definition was based on discretionary beverages. Fruit-

based smoothies were considered discretionary if they were store-bought, but homemade smoothies were assumed to be mainly comprised of whole fruits and milk or alternatives. Pure 100% fruit juices and flavoured milks (e.g., chocolate milk) were not classified as discretionary beverages as per the Australian Guide to Healthy Eating [23], due to their higher nutrient content [24]. Formulated supplementary foods and formulated sports foods (e.g., protein powdered drinks, ensure plus) were also excluded as were alcoholic beverages like beer and wine.

Data were stratified by: (1) age group, (2) gender, (3) consumers versus non-consumers of discretionary beverages and (4) drink categories. Nutritional information (daily intake of energy, protein, carbohydrates (CHO), total sugar, total fat, saturated fat and sodium) was determined for each participant. Total consumption and consumption by drink category were calculated by gender and age group.

*2.4. Statistical Analyses*

As the data were symmetrically distributed, parametric tests were employed. The proportions of consumers per drink category were compared by gender and by age group using the Z-test for two proportions. The average proportion of energy and nutrients contributed by discretionary beverages (per capita and per consumer) were calculated. Analysis of Covariance (ANCOVA) tests were used to compare energy and nutrient contributions (total energy, percent energy (%E) contributed by macronutrients and sodium density per 1000 kJ) between consumers and non-consumers adjusted for age group and gender. The adjusted means (estimated marginal means) were calculated using IBM SPSS statistics software (version 25, IBM, Chicago, IL, USA) with $p$-values < 0.05 considered statistically significant.

**3. Results**

*3.1. Overview*

Of the 1044 young adults enrolled in the study, 1001 (95.9%) contributed to the final analytical study (see Figure 1). Fifty-seven percent were females, 54% were aged 18–24 years, 52% had a university degree, 58% were from higher socioeconomic areas, 66% were from metropolitan areas and mean BMI was 25.7 kg/m$^2$. Among the 1001 participants, 616 (62%) were classified as consumers (participants who consumed at least one discretionary beverage over the three-day period), and 385 participants (38%) were non-consumers.

*3.2. Proportion of Discretionary Beverage Intake*

Table 1 presents the percentage of consumers of the discretionary beverage types by gender and age ($n$ = 616). The most commonly consumed discretionary beverages were soft drinks (39.0%, $n$ = 390) and flavoured teas/coffees (23.1%, $n$ = 231) for the total population. Soft drinks were the most commonly consumed discretionary beverage amongst both females (33.6%, $n$ = 190) and males (46.0%, $n$ = 200); and by age group i.e., 18–24-year-olds (37.8%, $n$ = 204) and 25–30-year-olds (40.3%, $n$ = 186). The second most popular discretionary beverage was flavoured tea/coffees for females (26.0%, $n$ = 147), and males (19.3%, $n$ = 84), and among age groups 18–24-year-olds (23.0%, $n$ = 124) and 25–30-year-olds (23.2%, $n$ = 107). A higher proportion of males consumed soft drinks ($p$ < 0.001) and energy/sport drinks than females ($p$ = 0.003) and a higher percentage of females than males consumed flavoured teas and coffees ($p$ = 0.013).

*3.3. Energy and Nutrient Contributed by Discretionary Beverages*

Table 2 presents the mean proportion of energy (kJ) and nutrients (g) contributed by discretionary beverages. The greatest proportion of nutrients contributed by discretionary beverages was from total sugars. Per capita, 16.6% of total sugar (g) and 7.0% of CHO (g) were contributed by discretionary beverage intake. For consumers, discretionary beverages contributed 27.2% of total sugar (g) and 11.5% of CHO (g).

**Table 1.** Proportion of consumers by type of discretionary beverages, age and gender.

| Discretionary Beverage Type | | Consumers, *n* (Percentage) | | | | | |
|---|---|---|---|---|---|---|---|
| | | **Gender** | | | **Age Group (Years)** | | |
| | | **Female** *n* 566 | **Male** *n* 435 | ***p*-Value** | **18–24** *n* 539 | **25–30** *n* 462 | ***p*-Value** |
| Soft drinks | 390 (39.0) | 190 (33.6) | 200 (46.0) | <0.001 | 204 (37.8) | 186 (40.3) | 0.419 |
| Flavoured tea/coffee | 231 (23.1) | 147 (26.0) | 84 (19.3) | 0.013 | 124 (23.0) | 107 (23.2) | 0.940 |
| Sweetened milk drinks | 81 (8.1) | 48 (8.5) | 33 (7.6) | 0.605 | 43 (8.0) | 38 (8.2) | 0.908 |
| Energy/sport drinks | 62 (6.2) | 24 (4.2) | 38 (8.7) | 0.003 | 35 (6.5) | 27 (5.8) | 0.647 |
| Flavoured water/cordial | 57 (5.7) | 31 (5.5) | 26 (6.0) | 0.736 | 30 (5.6) | 27 (5.8) | 0.892 |
| Fruit drinks | 56 (5.6) | 31 (5.5) | 25 (5.7) | 0.891 | 28 (5.2) | 28 (6.0) | 0.582 |
| Fermented drinks | 18 (1.8) | 11 (1.9) | 7 (1.6) | 0.721 | 7 (1.3) | 11 (2.4) | 0.193 |
| Any | 616 (61.6) | 333 (58.8) | 283 (65.1) | 0.042 | 337 (62.5) | 279 (60.4) | 0.496 |

**Table 2.** Contribution of discretionary beverages to total energy and nutrient intake per capita and per consumer, mean (95% confidence interval).

| | % Contribution per Capita | % Contribution per Consumer |
|---|---|---|
| Energy (kJ) | 3.8 (3.45–4.15) | 6.2 (5.74–6.70) |
| Protein (g) | 1.6 (1.35–1.81) | 2.6 (2.23–2.95) |
| Total fat (g) | 1.5 (1.32–1.78) | 2.5 (2.17–2.91) |
| Saturated fat (g) | 2.5 (2.09–2.87) | 4.1 (3.47–4.64) |
| Carbohydrate (g) | 7.0 (6.36–7.64) | 11.5 (10.67–12.39) |
| Total sugar (g) | 16.6 (15.3–17.89) | 27.2 (25.55–28.85) |
| Sodium (mg) | 1.7 (1.53–1.87) | 2.8 (2.56–3.04) |

*3.4. Comparison between Consumers and Non-Consumers*

Table 3 compares the mean total energy, the energy contributed by macronutrients and subtypes and sodium density between consumers and non-consumers adjusted for age group and gender. For non-consumers, the mean intakes were 7294 kJ, 20.5%E protein, 12.0%E saturated fat, 40.5%E carbohydrate and 13.3%E total sugar. In comparison, consumers of discretionary beverages had significantly higher intakes of energy (8736 kJ), saturated fat (12.5%E), carbohydrate (43.0%) and total sugar (16.5%) but lower intakes of protein (18.5%E) ($p < 0.05$). There were no significant differences in total fat and sodium density intake between consumers and non-consumers.

**Table 3.** Comparison of energy, % energy (%E) from nutrient intakes (mean 95% CI) between consumers and non-consumers *.

| | Consumer (*n* = 616) | | Non-Consumer (*n* = 385) | | |
|---|---|---|---|---|---|
| | **Mean** | **95% CI** | **Mean** | **95% CI** | ***p*-Value** |
| Total energy (kJ) | 8736 | 8510–8963 | 7294 | 7007–7581 | <0.001 |
| Protein (%E) | 18.5 | 18.1–18.9 | 20.5 | 20.0–21.0 | <0.001 |
| Total fat (%E) | 35.0 | 34.5–35.6 | 35.5 | 34.8–36.2 | 0.35 |
| Saturated fat (%E) | 12.5 | 12.2–12.8 | 12.0 | 11.6–12.4 | 0.024 |
| Carbohydrate (%E) | 43.0 | 42.3–43.6 | 40.5 | 39.7–41.4 | <0.001 |
| Total sugar (%E) | 16.5 | 16.0–17.1 | 13.3 | 12.7–14.0 | <0.001 |
| Sodium (mg/1000 kJ) | 320 | 312–329 | 316 | 305–327 | 0.511 |

* ANCOVA analysis adjusted for age group, gender, gender × age group.

## 4. Discussion

The findings of this cross-sectional study indicate that more than half (62%) of young adults, aged 18–30 years, consumed discretionary beverages at least once over a three-day period. These findings are similar to those of the NNPAS (*n* = 5396, 18–34-year-olds) and 2017–2018 NHS (*n* = 1998, 20–29-year-olds) surveys, where 54% of young adults consumed

discretionary beverages on the day of the survey and 56% of young adults consumed discretionary beverages at least once weekly [18,19]. On average, consumers' diets contained more than 1400 kJ additional energy per day than non-consumers and their total sugars were about 30 g more per day. According to a study that compares the 1995 NHS and 2011–2012 NNPAS, there has been a significant decrease in discretionary beverages contributing to added sugar consumption among all consumers (*p* < 0.001) [25]. Similarly, a decrease in the contribution of discretionary beverages to total sugar intake, when compared to the 2011–2012 NNPAS, 19.8% and 16.6% per capita, respectively, was revealed [26]. However, discretionary beverages remain one of the largest contributors to sugar consumption for both males and females up to the age of 70 years [25]. Furthermore, discretionary beverages have been linked with several health burdens such as being overweight and obesity [1,2,6,8–15,27].

Soft drinks were the most consumed discretionary beverages among all consumers. These results align with the 2011–2012 NNPAS, which reported soft drinks/flavoured mineral water as the most commonly consumed discretionary beverage amongst 19–30-year-olds [19]. There were observable differences between males and females in beverage preferences but not between 18- to 24-year-olds versus 25- to 30-year-olds in beverage consumption. Males were more likely to consume soft drinks and energy/sport drinks, whereas females were more likely to consume flavoured tea/coffee. These findings are paralleled in another study conducted in NSW in 2012 [12]. However, sweetened tea/coffee consumption was not significantly different between males and females in that study [12]. This could be due to the categorisation of discretionary beverages differing between the two studies or differences in preferences over time. The former study included milk coffees in the sweetened milk beverages category, whereas this study incorporates it into the flavoured tea/coffee category [12].

In another study of 675 18 to 30-year-old Australians, SSB consumption was found to be higher among males although the overall number of consumers among the population was only 35% over a 3 to 4-day period [28]. This result is anomalous in comparison to results found in 2011–2012 NNPAS (54%), 2017–2018 NHS (56%) and this study (62%). This is potentially because the majority of participants (58.8%, *n* = 397) were tertiary students with eating habits dissimilar to other young adults, and the generalisability of that study is limited [28].

Analysis of the UK National household survey that combined data from 2008 and 2009 to 2011 and 2012 also found differences in macronutrient quality between consumers of sugar sweetened and low-calorie beverages and non-consumers [29]. Consumers had a significantly higher intake of energy and a greater energy contribution from CHO and non-milk extrinsic sugar (*p* < 0.001) and had a lower contribution of energy from protein (*p* < 0.001) when compared with non-consumers [29]. As in the current study, total fat intake between non-consumers and consumers was not significantly different, but, unlike the current study, no increases in saturated fat %E were found [29]. The UK study found sodium intake to be higher amongst consumers of both sugar-sweetened and intensely sweetened beverages in comparison to non-consumers, 2559 mg/day and 2112 mg/day, respectively [29]. Some differences in observations may be the result of our study focusing exclusively on young adults, whereas the UK analysis was adults aged 19 to 64 years and there were differences in the classification of beverages.

A study using the National Health and Nutrition examination (NHANES) data from 2009–2014 exclusively focused on SSB consumption in adolescents (12 to 18 years) and young adults (19 to 29 years) and diet quality as assessed according to the 2015 Healthy Eating Index [30]. Diet quality of the food and beverage consumption was compared with and without the contribution of SSB. It was reported that non-consumers had a higher HEI score (Healthy Eating Index) compared with consumers (excluding SSB). Thus, removal of SSB from the diet would improve HEI, but other dietary changes are still needed for the remainder of their diets to improve overall scores for diet quality. The data used in the current study were collected for the MYMeals study that aimed to understand the

macronutrient consumption and intakes of deleterious nutrients of foods being sourced outside their homes. Interestingly, despite the high consumption of discretionary beverages, the mean CHO intake in the population was below the acceptable macronutrient range of 45 to 65%E, the total fat above the acceptable range of 20 to 35%E and the sodium density also above recommended levels [20,31]. This indicates that improving the diets of Australian young adults, similar to those in the US, will require public health measures involving a whole dietary pattern approach. Furthermore, different settings should be considered for intervention; for example, a previous study showed that young adults were consuming discretionary beverages during transport, with discretionary beverages being the largest contributor [32]. Substitution of discretionary beverages with water, unsweetened tea and coffee, and milk may be beneficial to lower weight gain [33–36]. In addition, higher water intakes are associated with better diet quality [37].

There are some limitations of the current study. Participants reported their own dietary intake, which may lead to misreporting due to social desirability and recall bias [38]. When the energy intake–calculated basal metabolic rate was considered, we were able to estimate that as many as 36% of this sample were underreporting their energy intake albeit 55% reported trying to lose weight during the study period [20]. Adjusting for energy intakes (such as reporting nutrients as a percentage of total energy) has been shown to reduce measurement error as the error in energy reporting is correlated with error in the reported intakes of all foods and beverages [20,39]. At the time of the study, the database had no data on added sugars only total sugars. However, one of the strengths of this study is that meaningful data have been analysed to provide a more updated insight into non-alcoholic discretionary beverage intake. Since the publication of the 2011–2012 NNPAS, a comprehensive study on beverage intake amongst young adults had not been reported until now. Therefore, the findings in this report will be informative for the praxis of health professionals and individuals with nutritional related health concerns. The study's focus on macronutrients' diet quality between consumers and non-consumers, 18–30 years old, adds new knowledge.

## 5. Conclusions

In conclusion, this report provides findings on the most recent comprehensive study on discretionary beverages amongst young adults in Australia. Australian young adults, 18–30 years old, continue to be high consumers of discretionary beverages, with soft drinks and flavoured teas and coffees the most consumed. There were observable differences between males and females in beverage preferences. Males had a higher consumption of soft drinks and energy/sport drinks, whereas females consumed more flavoured tea/coffee. Discretionary beverages contribute to a high proportion of energy and sugar, with macronutrient profiles differing between consumers and non-consumers, creating a less favourable profile. Among a range of measures to improve dietary intakes of young adults, beverage consumption is among the factors that should be prioritised as it has been linked to the prevalence of several health burdens.

**Author Contributions:** Conceptualization, M.A.-F.; methodology, M.A.-F., A.R. and L.W.-C.; formal analysis, A.D., P.Y.H., M.A.-F. and A.R.; data curation, A.J.D. and V.C.; writing—original draft preparation, A.D.; writing—review and editing, A.D., P.Y.H., M.A.-F., A.R., V.C., A.J.D. and L.W.-C.; visualization, funding acquisition, M.A.-F. and A.R. All authors have read and agreed to the published version of the manuscript.

**Funding:** This research uses data from a study funded by a Linkage Grant from the Australian Research Council and Cancer Council NSW LP150100831. L.W.-C. is employed by Cancer Council NSW, which co-funded the grant. A.J.D. and V.C. were funded by the Australian Government research training fund Ph.D. scholarship.

**Institutional Review Board Statement:** The study was conducted in accordance with the Declaration of Helsinki, and approved by the Institutional Review Board (or Ethics Committee) of University of Sydney Human Research Ethics Committee (protocol 2016/546).

**Informed Consent Statement:** Informed consent was obtained from all subjects involved in the study.

**Data Availability Statement:** Data sharing is not available for this study.

**Acknowledgments:** We would like to acknowledge all research members of the MYMeals study for the collection of the data and Luke Newby for his assistance with Excel and SPSS statistics.

**Conflicts of Interest:** M.A.-F. and A.R. declare other research funding from the National Health Medical Research Council and M.A.-F. from NSW Health. The funders had no role in the design of the study; in the collection, analyses, or interpretation of data; in the writing of the manuscript; or in the decision to publish the result. Other authors have no conflict of interest to declare.

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
