# Peer review of "High Consumption of Discretionary Beverages in Young Australian Adults Aged 18–30 Years: A Cross-Sectional Study"

_2674-0311, doi:10.3390/dietetics1020011_

Round 1

Reviewer 1 Report

Please refer to attached file.

Reviewer 2 Report

Thank you very much for allowing me to review the article entitled “High Consumption of Discretionary Beverages in Young Australian Adults aged 18-30 years: A Cross-Sectional Study” (dietetics-1764560). The aim of this study was to provide updated evidence for 18 to 30-year-olds in 2017/18 residing in New South Wales, Australia, regarding non-alcoholic discretionary beverage consumption including of sweetened water-based beverage and milk-based-beverage that are sweetened and may have added syrups and fats such as cream. And comparing the intake by sex of sources of sweetened beverage. This objective of having updated data I think should be modified, since the study belongs to the year 2017-18, we are already in 2022, that is, it is at least 4 years ago. The summary is informative and allows the reader to have an idea of ​​the content of the work in advance. The results that are exposed are interesting. The introduction is well thought out, the bibliography is adequate and up-to-date and fic justify the objective of the work. material and methods: A "cross-sectional" design is used, within the MYMeals study. They present the approval of the research ethics committee. The food intake is collected through a program using the smartphone app. A categorization of the drinks consumed is made into 7 groups and the young people are analyzed according to age, gender, consumption and non-consumption of drinks, and all this according to the category of the drink. It would be useful to have a flowchart to identify the study design and the participation of young people, it would allow a better understanding of the methodology. It should be justified why the age is divided into two groups from 18 to 24 and from 25 to 30. Results: They are presented in the form of a table, but the tables are not analyzed, a contrast test should be included between the different prevalence of beverage consumption among the different genders and different age groups. the lack of comparison using text confidence intervals in all the tables means that the results cannot be properly interpreted. Only table 3 presents an analysis of the information, but this does not take into account age or gender, therefore it does not correspond to the stated objective. Discussion: the discussion presents difficulties since statistical tests of homogeneity and contrast have not been applied in the population studied, it is difficult to interpret and compare the results with previous literature. I suggest that the confidence interval be calculated for the rate of consumption of each of these beverages, which will allow us to stabilize the result and the authors' abuse will allow it to be compared with studies on this subject. The conclusion presented in the manuscript does not respond to all the stated objectives.

Round 2

Reviewer 2 Report

Thank you very much for allowing me to review the manuscript again.

The authors' response to the comments is reasonable and they have mostly incorporated the suggestions.

I consider this to be an informative article on the reality of drinking among young Australians.